# Solving (partial) unbalanced optimal transport via transform coefficients and beyond

## Abstract

Unbalanced Optimal Transport (UOT) has gained increasing attention due to its ability to relax marginal constraints, thereby expanding its application potential. Previous solvers often incorporate an entropy regularization term, which can result in dense matching solutions. Meanwhile directly modeling UOT using penalized linear regression can be computationally expensive. To address the above issue, we turn to consider determining the marginal probability distribution of UOT with KL divergence via proposed *transform coefficient* method. The transform coefficient approach is not only computationally friendly but also reveals the essence of UOT, which involves adjusting the sample weights accordingly. We further extend the transform coefficient method into exploiting the marginal probability distribution of Partial Unbalanced Optimal Transport (PUOT) with KL divergence for validating its generalization. Since the marginal probability of UOT/PUOT are determined, we are excited to discover that UOT/PUOT can be transformed into classical Optimal Transport (OT) problem for finding the transportation plan. Therefore, the transform coefficient method can be considered as the bridge that establishes the connection between UOT/PUOT and OT. Moreover, we discover the additional results of Lagrange multipliers when solving transform coefficient can offer valuable guidance for achieving more sparse and accurate mapping with Cost-Reweighted OT (CROT). We perform several numerical experiments to illustrate our proposed new algorithms on dealing with UOT, PUOT and OT problem.

## 1 Introduction

Optimal Transport (OT) technique is a powerful tool for discerning and comparing distinct probability distributions. Nowadays, OT has multiple successful applications in traditional machine learning (Frogner et al., 2015; Feydy et al., 2019; Zhuang et al., 2022; Chuang et al., 2023), unsupervised clustering (Asano et al., 2019; Caron et al., 2020), domain adaptation (Damodaran et al., 2018; Courty et al., 2017; Redko et al., 2019), model diffusion modelling (Khrulkov et al., 2023; Lipman et al., 2023), generative modelling (Korotin et al., 2023; Onken et al., 2021; Tong et al., 2023) and many others. Nevertheless, directly solving OT distances could have relatively high computation cost with around super-cubic time. Although one can adopt entropy-based Sinkhorn algorithm (Cuturi, 2013) for solving OT efficiently, it still suffers from the dilemma of dense solution problem Liu et al. (2023); Lorenz et al. (2021); Dessein et al. (2018). Moreover, classical OT strictly assume that the masses on both source and target domains should be equal. It further hurdles the generalization of OT for tackling the scenario when the data samples inherit noise or outliers.

Recently, Unbalanced Optimal Transport (UOT) (Benamou, 2003; Chizat, 2017) technique has become more attractive since it allows mass variation during solving the transportation results. UOT adopts several divergences such as Kullback-Leiber (KL) divergence (Pham et al., 2020), $\ell_1$ norm (Caffarelli & McCann, 2010) and $\ell_2$ norm (Blondel et al., 2018; Chapel et al., 2021) for the relaxation on OT mass equality constraints. Meanwhile KL divergence is the most commonly-used in UOT formulation in real practice Séjourné et al. (2022). UOT also provides great applications in transfer learning (Fatras et al., 2021; Tran et al., 2023; Mukherjee et al., 2021), computer vision (Bonneel & Coeurjolly, 2019), structure data exploration (Sato et al., 2020), natural language processing (Arase et al., 2023) and many areas. Previous solvers always involves some regularization terms including entropy regularization term (Fatras et al., 2021) and proximal point term (Chapel et al., 2021) for tackling UOT problem. While adding additional entropy terms will lead to dense and

inaccurate solution of matching. Latest, (Chapel et al., 2021) further reconsiders UOT problem as penalized linear regression without any entropy regularization term. Thus UOT can achieve sparse and accurate solution via regularization path method (Chapel et al., 2021). While regularization path method should utilize matrix inversion progressively results in high space-time consumption.

In this paper, we propose a new method, i.e., ***transform coefficients***, for solving UOT with KL divergence without entropy regularization term. Since it is difficult to directly obtain the matching results of UOT, we originally turn to consider whether it is possible to first obtain the marginal probability distribution of the solution. To do so, we make calculations based on the KKT conditions of UOT, finding the proposed transform coefficients for determining the marginal probability distributions. We can further observe that the essence of UOT lies in adjusting the initial weights of different data samples accordingly, which provides us with new insights for understanding UOT. Moreover, we observe that any UOT with KL divergence can be equivalently represented as the corresponding OT problem through transform coefficients. As we already have abundant methods available for tackling OT problems, the idea of converting UOT to OT brings us a brand new perspective in dealing with UOT. Next we expand our approach into solving Partial Unbalanced Optimal Transport (PUOT) with KL divergence and successfully transform PUOT into OT as expected. Moreover, we discover that while solving the marginal distribution, the Lagrange multipliers can offer valuable guidance for addressing the OT problem. Therefore we further proposed Cost-Reweighted Optimal Transport (CROT) for achieving more sparse and accurate OT matching solution.

## 2 PRELIMINARY

To start with, we first provide a brief preliminary definition of OT, UOT and PUOT. Let us consider two sets of data samples $\boldsymbol{X} \in \mathbb{R}^{M \times D}$ and $\boldsymbol{Z} \in \mathbb{R}^{N \times D}$ in source and target domains, where $M$, $N$ denote the number of samples and $D$ denotes the data dimension. Each data samples has corresponding mass $\boldsymbol{a} \in \mathbb{R}^{M \times 1}$ and $\boldsymbol{b} \in \mathbb{R}^{N \times 1}$. Meanwhile the total masses of these data samples are equal as $\boldsymbol{a}^\top \mathbf{1}_M = \boldsymbol{b}^\top \mathbf{1}_N$. The classical OT problem was defined by (Kantorovich, 1942) with a linear problem to measure the minimum transportation cost from moving data sample $\boldsymbol{X}$ to $\boldsymbol{Z}$:

$$\text{OT}(\boldsymbol{a}, \boldsymbol{b}) = \min_{\pi_{ij} \geq 0} \langle \boldsymbol{C}, \boldsymbol{\pi} \rangle \quad \text{s.t. } \boldsymbol{\pi} \mathbf{1}_N = \boldsymbol{a}, \quad \boldsymbol{\pi}^\top \mathbf{1}_M = \boldsymbol{b}, \tag{1}$$

where $\boldsymbol{C} \in \mathbb{R}^{M \times N}$ denotes the pairwise distance which can be calculated via $C_{ij} = ||\boldsymbol{X}_i - \boldsymbol{Z}_j||_2^2$. Meanwhile $\boldsymbol{\pi} \in \mathbb{R}^{M \times N}$ denotes the coupling matching matrix among the data samples $\boldsymbol{X}$ and $\boldsymbol{Z}$. One can directly solve equation 1 via utilizing network-flow algorithm (Kennington & Helgason, 1980; Ahuja et al., 1988). To consider more general cases (e.g., filtering out the noise or outliers), one can relax two marginal constraints to achieve unbalanced optimal transport problem:

$$\text{UOT}^\tau(\boldsymbol{a}, \boldsymbol{b}) = \min_{\pi_{ij} \geq 0} \quad \langle \boldsymbol{C}, \boldsymbol{\pi} \rangle + \tau_a \cdot \text{KL}(\boldsymbol{\pi} \mathbf{1}_N \| \boldsymbol{a}) + \tau_b \cdot \text{KL}(\boldsymbol{\pi}^\top \mathbf{1}_M \| \boldsymbol{b}), \tag{2}$$

where $\text{KL}(\cdot)$ denotes Kullback-Leiber (KL) divergence which has been widely used in dealing with UOT (). $\tau_a$ and $\tau_b$ denote the balanced hyper parameters between the minimizing cost and marginal relaxation. For simplification, we set $\tau_a = \tau_b = \tau$ in the following discussion. Note that when $\tau \to +\infty$ and $\boldsymbol{a}^\top \mathbf{1}_M = \boldsymbol{b}^\top \mathbf{1}_N$, UOT problem will turn into classical OT. Moreover, we can only relax one marginal constraints to formulate PUOT. For instance, we relax the constraint $\boldsymbol{\pi} \mathbf{1}_N = \boldsymbol{a}$ while keep the constraint $\boldsymbol{\pi}^\top \mathbf{1}_M = \boldsymbol{b}$:

$$\text{PUOT}^\tau(\boldsymbol{a}, \boldsymbol{b}) = \min_{\pi_{ij} \geq 0} \quad \langle \boldsymbol{C}, \boldsymbol{\pi} \rangle + \tau \cdot \text{KL}(\boldsymbol{\pi} \mathbf{1}_N \| \boldsymbol{a}) \quad \text{s.t. } \boldsymbol{\pi}^\top \mathbf{1}_M = \boldsymbol{b}. \tag{3}$$

Previous researches always add entropy regularization term for solving OT, UOT and PUOT. Although entropy regularization term can enhance the scalability of solving $\boldsymbol{\pi}^*$, it still suffers from the dense solution dilemma which inaccurate results. In this following paper, we will first investigate the problem of UOT/PUOT from the perspective of marginal probability distribution, trying to find out the sparse and accurate solution of $\boldsymbol{\pi}^*$ for OT, UOT and PUOT.

## 3 METHODOLOGY

In this section, we will provide the calculation details on finding the solutions for commonly-existed UOT and PUOT. Previous methods (Pham et al., 2020; Fatras et al., 2021) always directly adopted entropy-based regularization term into tackling UOT and PUOT problem. Although such approaches can provide fast computation speed, it will lead to relatively dense solution which does not match most of situations in real practices. Latest, Chapel et al. (2021) adopted

majorization-minimization algorithm or regularization path for solving UOT/PUOT problem. However, majorization-minimization algorithm could be inaccurate when $\tau \to +\infty$ since it will provide dense solutions. Meanwhile, regularization path could involve heavy matrix computation on inversion with complicated optimization procedure. To better solve the above problem, we change the perspective of reviewing the UOT/PUOT problem, that is, not to directly solve the coupling matrix $\boldsymbol{\pi}$, but pay attention to exploiting the marginal probability of UOT/PUOT. From that we can obtain some interesting insights on understanding the relationship between UOT/PUOT and classical optimal transport. Moreover, these new proposed theorems and corollaries can help to achieve more accurate the sparse matching solution efficiently.

## 3.1 FINDING MARGINAL PROBABILITY FOR UOT

To start with, let we first exploit the marginal probability for UOT. To better fulfill this task, we need to involve newly proposed transform coefficients for calculation. Meanwhile, we aim not to solve matching matrix $\boldsymbol{\pi}$ directly during the optimization process for avoid heavy computation. Then we introduce the calculation details as follows:

***Proposition 1.*** *Given any UOT problem with KL divergence, the marginal probability can be determined without calculating specific solution on matrix $\boldsymbol{\pi}^*$ as:*

$$\sum_{i=1}^{M} \pi_{ij} = \frac{b_j \psi_j}{\sqrt{\sum_{p=1}^{N} b_p \psi_p}} = \beta_j \quad \text{and} \quad \sum_{j=1}^{N} \pi_{ij} = \frac{a_i \delta_i}{\sqrt{\sum_{q=1}^{M} a_q \delta_q}} = \alpha_i \tag{4}$$

*where $\boldsymbol{\psi}$ and $\boldsymbol{\delta}$ are denoted as UOT transform coefficients and they can be calculated via $\psi_j = \sum_{i=1}^{M} a_i \exp(-\widehat{C}_{i,j}^*/\tau)$ and $\delta_i = \sum_{j=1}^{N} b_j \exp(-\widehat{C}_{i,j}^*/\tau)$ with transformed pairwise distance $\widehat{C}_{i,j}^*$.*

We illustrate Proposition 1 via providing details on optimizing the marginal probability for UOT.

**KKT conditions of UOT.** We first consider the Lagrange multipliers of UOT with KL divergence:

$$L_{\mathrm{UOT}} = \max_{\boldsymbol{s} \geq 0} \min_{\pi_{ij} \geq 0} \langle \boldsymbol{C}, \boldsymbol{\pi} \rangle + \tau \cdot \mathrm{KL}\left(\boldsymbol{\pi} \mathbf{1}_N \| \boldsymbol{a}\right) + \tau \cdot \mathrm{KL}(\boldsymbol{\pi}^\top \mathbf{1}_M \| \boldsymbol{b}) - \langle \boldsymbol{s}, \boldsymbol{\pi} \rangle \tag{5}$$

where $\boldsymbol{s}$ denotes Lagrange multipliers. KKT optimal conditions illustrate that (1) $\boldsymbol{s} \odot \boldsymbol{\pi} = 0$ (complementary condition), (2) $s_{ij} \geq 0$ (feasibility condition), (3) $\nabla_{\pi_{ij}} L_{\mathrm{UOT}} = C_{ij} + \tau \log((\boldsymbol{\pi} \mathbf{1}_N)_i / a_i) + \tau \log((\boldsymbol{\pi}^\top \mathbf{1}_M)_j / b_j) - s_{ij} = 0$ (stationary condition).

**Determining Transform Coefficients of UOT.** Secondly, we should calculate the transform coefficients of UOT. To facilitate the following computation, we let $\tau \log((\boldsymbol{\pi} \mathbf{1}_N)_i / a_i) = -u_i$ and $\tau \log((\boldsymbol{\pi}^\top \mathbf{1}_M)_j / b_j) = -v_j$ with the corresponding mass-equality and KKT stationary equations:

$$\sum_{j=1}^{N} b_j \exp\left(-\frac{v_j}{\tau}\right) = \sum_{i=1}^{M} a_i \exp\left(-\frac{u_i}{\tau}\right) \quad \text{and} \quad \widehat{C}_{ij} = C_{ij} - s_{ij} = u_i + v_j. \tag{6}$$

Meanwhile we should figure out the unknown values of $\boldsymbol{u}$, $\boldsymbol{v}$ and $\boldsymbol{s}$ while avoid solving $\boldsymbol{\pi}$. We set $s_{ij}^{(0)} = 0$ for initialization and figuring out the value of $\boldsymbol{v}$ at the $l$-th iteration as:

$$\sum_{j=1}^{N} b_j \exp\left(-\frac{v_j^{(l)}}{\tau}\right) = \exp\left(\frac{v_w^{(l)}}{\tau}\right) \sum_{i=1}^{M} a_i \exp\left(-\frac{\widehat{C}_{iw}^{(l)}}{\tau}\right) \quad \text{for} \quad \forall w \in \{1, 2, \cdots, N\}$$

Here we denote $\psi_w^{(l)} = \sum_{i=1}^{M} a_i \exp(-\widehat{C}_{i,w}^{(l)}/\tau)$ as one of the UOT transform coefficients for facilitating the calculation process and we could obtain the results of $\boldsymbol{v}^{(l)}$ via $v_j^{(l)} = -\tau \log(\psi_j^{(l)}/\Psi^{(l)})$ where $\Psi^{(l)} = \sqrt{\sum_{p=1}^{N} b_p \psi_p^{(l)}}$. Likewise, we can optimize another variable $\boldsymbol{u}$ using similar way:

$$\sum_{i=1}^{M} a_i \exp\left(-\frac{u_i^{(l)}}{\tau}\right) = \exp\left(\frac{u_r^{(l)}}{\tau}\right) \sum_{j=1}^{N} b_j \exp\left(-\frac{\widehat{C}_{rj}^{(l)}}{\tau}\right) \quad \text{for} \quad \forall r \in \{1, 2, \cdots, M\}$$

Then we denote $\delta_r^{(l)} = \sum_{j=1}^{N} b_j \exp(-\widehat{C}_{r,j}^{(l)}/\tau)$ as another UOT transform coefficients for achieving the results of $\boldsymbol{u}^{(l)}$ via $\boldsymbol{u}^{(l)}$ via $u_i^{(l)} = -\tau \log(\delta_i^{(l)}/\Delta^{(l)})$ where $\Delta^{(l)} = \sqrt{\sum_{q=1}^{M} a_q \delta_q^{(l)}}$. After the $l$-th iteration, we further optimize the value of $\boldsymbol{s}$ via $s_{ij}^{(l+1)} = \max(0, C_{ij} - u_i^{(l)} - v_j^{(l)})$ for the next iteration.

***Corollary 1 from Proposition 1.*** *Given any UOT problem with KL divergence, the marginal probability can be determined as $\sum_{i=1}^{M} \pi_{ij} = \frac{A}{\sqrt{AB}} b_j$ and $\sum_{j=1}^{N} \pi_{ij} = \frac{B}{\sqrt{AB}} a_i$ when $\tau$ approaches to the infinity ($\tau \to +\infty$) and A, B denotes the sum of initial sample weights ($A = \sum_{i=1}^{M} a_i$, $B = \sum_{j=1}^{N} b_j$).*

***Corollary 2 from Proposition 1.*** *Given any UOT problem with KL divergence and $\boldsymbol{a}^\top \mathbf{1}_M = \boldsymbol{b}^\top \mathbf{1}_N$, the marginal probability can be determined as $\sum_{i=1}^M \pi_{ij} = b_j$ and $\sum_{j=1}^N \pi_{ij} = a_i$ when $\tau$ approaches to the infinity ($\tau \to +\infty$) and at that time UOT is equivalent to classical OT problem.*

**Brief Summary.** Due to the above observations, the marginal probability for any UOT with KL divergence can be determined via our proposed transform coefficients. Apparently, the samples with small initial weights or far away from the others will easily have lower value of transform coefficients. Meanwhile different value of $\tau$ will also affect the weights of different data samples on marginal probability. We can also exploit some interesting corollaries from exploiting UOT transform coefficients. The whole computation procedures are given in Algorithm 1. Note that we don't need to obtain the optimal solution of coupling matrix $\boldsymbol{\pi}^*$ when calculating the marginal probability. What is more, we do not involve any regularization terms during optimization.

---

**Algorithm 1** The training procedure of transform coefficients on UOT with KL Divergence

---

**Input:** $C$: cost matrix; $\boldsymbol{a}, \boldsymbol{b}$: initial marginal probability; $\tau$: Hyper parameters.
Set $s_{ij}^{(0)} = 0$ as the initialization.
**for** $l = 1$ to $L$ **do**
  Obtain transformed pairwise distance $\widehat{C}_{ij}^{(l)} = C_{ij} - s_{ij}^{(l)}$.
  Obtain UOT transform coefficients: $\psi_j^{(l)} = \sum_{i=1}^M a_i \exp(-\frac{\widehat{C}_{i,j}^{(l)}}{\tau})$, $\delta_i^{(l)} = \sum_{j=1}^N b_j \exp(-\frac{\widehat{C}_{i,j}^{(l)}}{\tau})$.
  Obtain $v_j^{(l)} = -\tau \log(\psi_j^{(l)}/\Psi^{(l)})$, $u_i^{(l)} = -\tau \log(\delta_i^{(l)}/\Delta^{(l)})$ where $\Psi^{(l)} = \sqrt{\boldsymbol{b}^\top \boldsymbol{\psi}^{(l)}}$, $\Delta^{(l)} = \sqrt{\boldsymbol{a}^\top \boldsymbol{\delta}^{(l)}}$.
  Obtain multipliers via $s_{ij}^{(l+1)} = \max(0, C_{ij} - u_i^{(l)} - v_j^{(l)})$.
**end for**
**Return:** The UOT transform coefficients and $s_{ij}^{(L+1)}$.

---

### 3.2 Finding marginal probability for PUOT

In Section 3.1, we have obtained the transform coefficients to determine the marginal probability for UOT. The method even without directly solving the coupling matrix $\boldsymbol{\pi}^*$. In this section, we will further extend our methods for solving the marginal probability on PUOT, which is also a commonly exist optimization problem (Chapel et al., 2021; Le et al., 2021; Schiebinger et al., 2017).

***Proposition 2.*** *Given any PUOT with KL divergence, the marginal probability can be determined without calculating the specific solution on matrix $\boldsymbol{\pi}^*$ as:*

$$\sum_{j=1}^N \pi_{ij} = \Gamma_i \quad \text{and} \quad \Gamma_i = a_i \exp\left(\frac{1}{N\tau}\sum_{j=1}^N (g_j - \widehat{C}_{ij})\right) \tag{7}$$

*where $\boldsymbol{\Gamma}$ denotes PUOT transform coefficients and $\boldsymbol{g}$ denotes the multipliers which can be calculated as $g_j = \tau \log(\sum_{k=1}^N b_k) - \tau \log(\sum_{i=1}^M a_i \exp(-\widehat{C}_{ij}/\tau))$ with transformed pairwise distance $\widehat{C}_{i,j}$.*

We illustrate Proposition 2 via providing details on optimizing the marginal probability for PUOT.

**KKT conditions of PUOT.** We first consider the Lagrange multipliers of PUOT with KL divergence:

$$L_{\text{PUOT}} = \max_{\boldsymbol{s} \geq 0, \boldsymbol{g}} \min_{\pi_{ij} \geq 0} \langle \boldsymbol{C}, \boldsymbol{\pi} \rangle + \tau \cdot \text{KL}(\boldsymbol{\pi}\mathbf{1}_N \| \boldsymbol{a}) - \langle \boldsymbol{g}, \boldsymbol{\pi}^\top \mathbf{1}_M - \boldsymbol{b} \rangle - \langle \boldsymbol{s}, \boldsymbol{\pi} \rangle \tag{8}$$

where $\boldsymbol{s}$ and $\boldsymbol{g}$ denotes Lagrange multipliers. KKT optimal conditions illustrate that (1) $\boldsymbol{\pi}^\top \mathbf{1}_M = \boldsymbol{b}$ (boundary condition), (2) $\boldsymbol{s} \odot \boldsymbol{\pi} = 0$ (complementary condition), (3) $s_{ij} \geq 0$ (feasibility condition), (4) $\nabla_{\pi_{ij}} L_{\text{PUOT}} = C_{ij} + \tau \log((\boldsymbol{\pi}\mathbf{1}_N)_i/a_i) - g_j - s_{ij} = 0$ (stationary condition).

**Optimization on multipliers $\boldsymbol{g}$.** We first try to figure out the value of multipliers $\boldsymbol{g}$. Similar to the optimization in UOT, we let $\sum_{j=1}^N \pi_{ij} = a_i \exp(-f_i/\tau) = \Gamma_i$ to obtain the following equation $\widehat{C}_{ij} = C_{ij} - s_{ij} = f_i + g_j$. We also set $s_{ij}^{(0)} = 0$ in initialization for facilitating the calculation. It is obvious that $\sum_{i=1}^M \pi_{ij} = \sum_{j=1}^N b_j$ and we can expand this formula at the $l$-th iteration accordingly:

$$\sum_{j=1}^N b_j = \sum_{i=1}^M a_i \exp\left(\frac{g_u^{(l)} - \widehat{C}_{iu}^{(l)}}{\tau}\right) \Rightarrow g_u^{(l)} = \tau \log\left(\frac{\sum_{j=1}^N b_j}{\sum_{i=1}^M a_i \exp(-\widehat{C}_{iu}^{(l)}/\tau)}\right) \tag{9}$$

**Optimization on PUOT transform coefficients $\boldsymbol{\Gamma}$.** Since we have obtained the value of multipliers $\boldsymbol{g}^{(l)}$, we can achieve $\boldsymbol{f}^{(l)}$ via minimizing the following equation $[\min_{f_i^{(l)}} \sum_{j=1}^N \|\widehat{C}_{ij}^{(l)} - f_i^{(l)} - g_j^{(l)}\|_2^2]$.

The solution is clear to obtain as $\boldsymbol{f}^{(l)} = \sum_{j=1}^{N}(\widehat{C}_{ij}^{(l)} - g_j^{(l)})/N$ for the optimal estimation. Finally, we can obtain the PUOT transform coefficients $\boldsymbol{\Gamma}^{(l)}$ via $\boldsymbol{\Gamma}^{(l)} = a_i \exp(-\sum_{j=1}^{N}(\widehat{C}_{ij}^{(l)} - g_j^{(l)})/N\tau)$.

**Optimization on multipliers $\boldsymbol{s}$.** After we obtain multipliers $\boldsymbol{g}^{(l)}$ and PUOT transform coefficients $\boldsymbol{\Gamma}^{(l)}$, we can further optimize $\boldsymbol{s}$ via $s_{ij}^{(l+1)} = \max(0, C_{ij} - f_i^{(l)} - g_j^{(l)})$ for the next $(l+1)$-th iteration.

**Brief Summary.** Optimizing the marginal probability distribution on PUOT is rather similar to the process mentioned in Section 3.1, which indicates that our proposed transform coefficient method can be extended to more application scenarios. Specifically, we can figure out the marginal probability on PUOT without directly obtain the value of coupling matrix $\boldsymbol{\pi}^*$ during the optimization.

### 3.3 THEOREM APPLICATION ON FINDING MAPPING SOLUTION

According to the (Proposition 1, Proposition 2) that discussed in Section 3.1 and 3.2, we have figured out the marginal probability distributions on both UOT and PUOT with commonly used KL Divergence. From that we can exploit the core mechanism of UOT/PUOT is *carefully reweighted* the weights of different samples. If the samples are noise or outliers, the corresponding weights will be much smaller. Otherwise, the corresponding weights among similar data samples will become larger. Based on the reweighted mechanism, UOT/PUOT has better adaptability than traditional OT which commonly treat all data samples equally. Moreover, the KL Divergence terms (e.g., $\mathrm{KL}\left(\boldsymbol{\pi}\mathbf{1}_N\|\boldsymbol{a}\right)$ and $\mathrm{KL}\left(\boldsymbol{\pi}^\top\mathbf{1}_M\|\boldsymbol{b}\right)$ in UOT) become the constants. Therefore, we can obtain the following corollary among (UOT, PUOT) and OT as:

***Corollary 3.*** *Given any UOT/PUOT with KL divergence, we can transfer the original optimization problem into classical optimal transport via adopting newly proposed UOT/PUOT transform coefficients. We can further utilize existing OT solver for solving $\boldsymbol{\pi}^*$ of UOT/PUOT as:*

$$(\mathrm{UOT}, \mathrm{PUOT}) \xrightarrow{\mathrm{UOT/PUOT\ Transform\ Coefficients}} \mathrm{OT} \xrightarrow{\mathrm{OT\ Solver}} \boldsymbol{\pi}^* \qquad (10)$$

This observation brings us a complete new insights on solving the coupling matrix $\boldsymbol{\pi}^*$ for UOT and PUOT. The transformation via transform coefficient is meaningful, mainly because there exists much more efficient and accurate OT solvers than directly optimize UOT/PUOT. Specifically, one can adopt network-flow solver to calculate $\boldsymbol{\pi}^*$ with precise results with relatively high computation cost. Or one can adopt more efficient methods (e.g., Sinkhorn (Cuturi, 2013)) that involve some additional regularization terms to figure out $\boldsymbol{\pi}^*$ with relatively high speed.

However, previous efficient OT solvers always suffer from the dilemma that matching results are relatively dense. This is unreasonable since the solution of $\boldsymbol{\pi}^*$ should be sparse in most cases. Recalling the whole process of Proposition 1/Proposition 2, we not only obtain the marginal probability with transform coefficients, but also obtain the value of multipliers $\boldsymbol{s}$ which can be further utilized. According to the KKT complementary and feasibility conditions, when $s_{ij} = 0$ it leads to $\pi_{ij} > 0$, otherwise $s_{ij} > 0$ indicates that $\pi_{ij} = 0$. In other words, the value of $\pi_{ij}$ can be reflected via $s_{ij}$. It inspired us to further consider such useful information in calculating the optimal transport of $\boldsymbol{\pi}^*$.

***Proposition 3.*** *Given any OT with multiplier $\boldsymbol{s}$, one can obtain sparse mapping solution via multiplying the cost matrix $\boldsymbol{C}$ with multiplier $\boldsymbol{s}$ to form Cost-Reweighted Optimal Transport (CROT):*

$$\min_{\boldsymbol{\pi}}\langle\boldsymbol{C}\odot\eta\boldsymbol{s}, \boldsymbol{\pi}\rangle = \langle\widetilde{\boldsymbol{C}}, \boldsymbol{\pi}\rangle \quad s.t.\ \boldsymbol{\pi}\mathbf{1}_N = \widehat{\boldsymbol{\alpha}}, \quad \boldsymbol{\pi}^\top\mathbf{1}_M = \widehat{\boldsymbol{\beta}}, \quad \pi_{ij} \geq 0 \qquad (11)$$

*where $\widetilde{\boldsymbol{C}}$ denotes the reweighted cost matrix and $\eta$ represents a sufficiently large positive number.*

Apparently, we multiply cost matrix $\boldsymbol{C}$ with multiplier matrix $\boldsymbol{s}$ to form a new reweighted cost matrix $\widetilde{\boldsymbol{C}}$. Therefore, the cost will be re-scaled according to the value of corresponding multipliers. If $s_{ij}$ is much approaches to 0, it indicates that it has higher probability to be matched in $\pi_{ij}$ and it is reasonable to reduce the cost distance between them. Otherwise, we should extend their distances by further multiplying $s_{ij}$ with a sufficiently large positive scalar $\eta$, to avoid the pairwise matching. Thus, the useful information of multipliers can be introduced into optimal transport, making the distances become more discriminative for achieving sparse and accurate solution on $\boldsymbol{\pi}^*$.

***Corollary 4.*** *Given any OT, one can first choose large value of $\tau$ to transfer it into UOT/PUOT problem for finding the multipliers $\boldsymbol{s}$, then establishing CROT for sparse matching solution as:*

$$\mathrm{OT} \xrightarrow{\mathrm{Large}\ \tau} (\mathrm{UOT}, \mathrm{PUOT}) \xrightarrow{\mathrm{Multipliers}\ \boldsymbol{s}} \mathrm{Reweighted\ OT} \xrightarrow{\mathrm{OT\ Solver\ (e.g., Sinkhorn)}} \boldsymbol{\pi}^* \qquad (12)$$

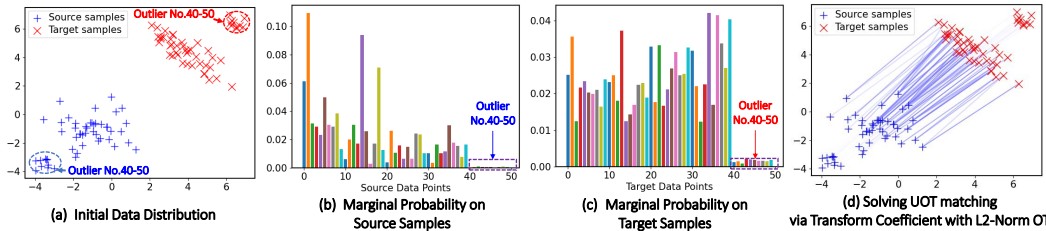

Figure 1: The marginal probabilities among source/target samples and matching solution on UOT.

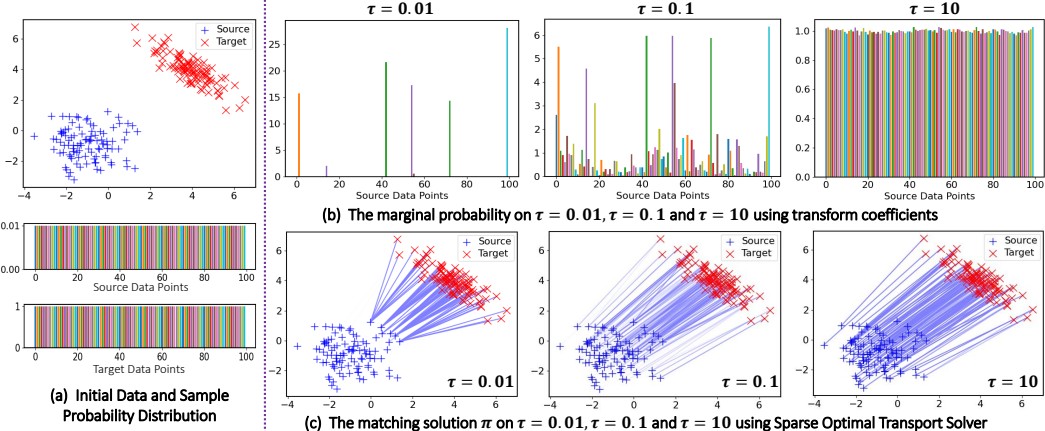

Figure 2: The marginal probability among source/target samples and matching solution on PUOT.

This observations bring us another new insight on considering the OT problem. Specifically, transform coefficient method not only provides the marginal probability information among UOT/PUOT, but also serves as the connection bridge between UOT/PUOT and classical OT.

## 4 NUMERICAL EXPERIMENTS

In this section, we will show the obtained solutions on simple and interpretable examples for validation. To start with, we first provide the solutions on finding marginal probability of UOT/PUOT using proposed transform coefficients method. Then we adopt traditional OT solvers to obtain the matching results $\pi^*$ for UOT/PUOT and evaluating the computation cost. Finally, we investigate the multipliers to establish cost-reweighted optimal transport with other OT solvers.

**Visualization on Marginal Probability of UOT/PUOT.** We first illustrate the learned marginal probability of given UOT. Following previous works (Flamary et al., 2021; Chapel et al., 2021), we sample 40 points to build up the source and target domains from $\boldsymbol{x}^+ \sim \mathbb{P}_X$, $\boldsymbol{z}^+ \sim \mathbb{P}_Z$ where $\mathbb{P}_X = \mathcal{N}\left(\begin{bmatrix} -1 \\ -1 \end{bmatrix}, \begin{bmatrix} 1 & 0 \\ 0 & 1 \end{bmatrix}\right)$ and $\mathbb{P}_Z = \mathcal{N}\left(\begin{bmatrix} 4 \\ 4 \end{bmatrix}, \begin{bmatrix} 1 & -0.8 \\ -0.8 & 1 \end{bmatrix}\right)$ respectively. Then we further sample 10 outliers from $\boldsymbol{x}^- \sim U\left([-4, -3] \times [-4, -3]\right)$ and $\boldsymbol{z}^- \sim U\left([6, 7] \times [6, 7]\right)$ for source and target domains and denote them from No.40 to No.50. The initial data distribution has been shown in Fig. 1(a) where we depict source and target samples (i.e., $\boldsymbol{x} = [\boldsymbol{x}^+; \boldsymbol{x}^-]$ and $\boldsymbol{z} = [\boldsymbol{z}^+; \boldsymbol{z}^-]$) with blue and red colors respectively. We adopt square Euclidean distance to measure the cost via $C_{ij} = ||\boldsymbol{x}_i - \boldsymbol{z}_j||_2^2$ then divide by $\max_{ij}[C_{ij}]$ for normalization and set $\tau = 0.05$, $a_i = \frac{1}{50}$ and $b_j = \frac{1}{50}$ following (Flamary et al., 2021; Chapel et al., 2021). Then we utilize our proposed transform coefficient method to figure out the marginal probability (i.e., the value of $\boldsymbol{\alpha}$ and $\boldsymbol{\beta}$ in equation 4) for both source and target domains via Algorithm 1 and the results have been shown in Fig. 1(b)-(c). We can observe that these outliers (data samples No.40-No.50) have much lower marginal probabilities among the data samples which indicates that the essence of UOT is to reweight different samples accordingly. Meanwhile, we should validate the performance of transform coefficients on PUOT scenario. Following previous works (Flamary et al., 2021; Chapel et al., 2021), we sample 100 points to form source and target domains as $\boldsymbol{x} \sim \mathbb{P}_X$ and $\boldsymbol{z} \sim \mathbb{P}_Z$ respectively. To simulate the scenario when the mass are different across domains, we set $a_i = \frac{1}{100}$ and $b_j = 1$ as shown in Fig. 2(a). We also adopt proposed transform coefficient method to figure out the marginal probability (i.e., the value of transformation coefficient $\boldsymbol{\Gamma}$ in equation 7) when choosing different value of $\tau$ as $\tau \in \{0.01, 0.1, 10\}$. The results are shown in Fig. 2(b) and we can observe that when $\tau$ is small,

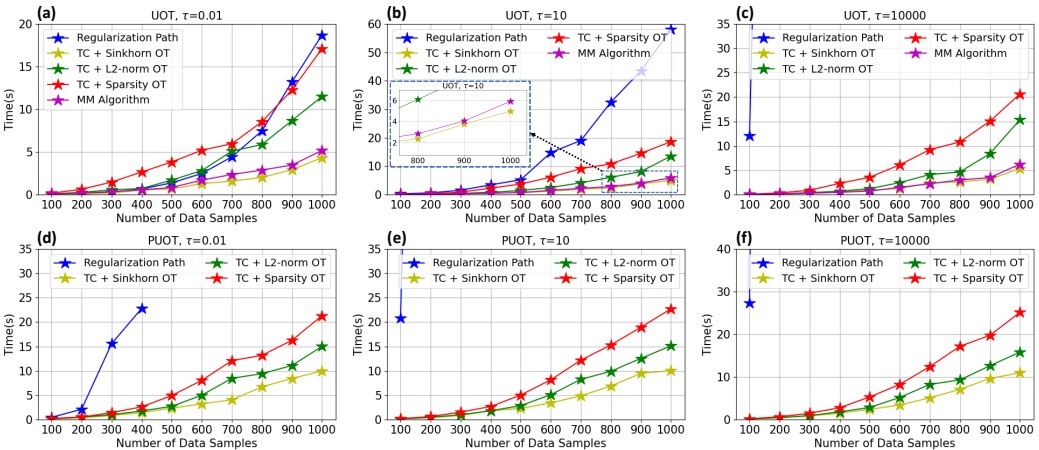

Figure 3: Time consumption analysis on solving UOT/PUOT with different value of $\tau$. Here we denote transform coefficient as TC for simplification in the legend. Note that MM Algorithm cannot be directly applied in solving PUOT Chapel et al. (2021).

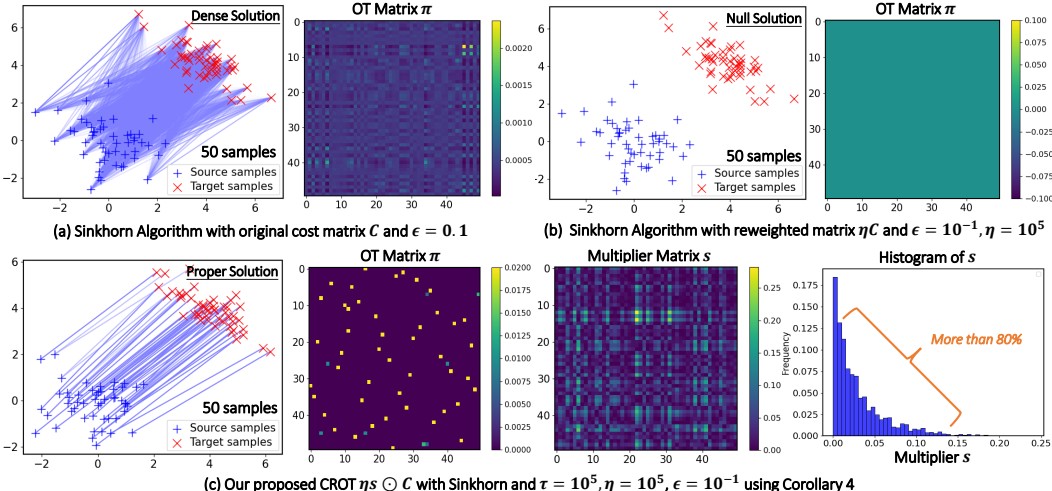

Figure 4: The visualization on original Sinkhorn and our proposed CROT with Sinkhorn using 2D empirical distributions.

only relatively few source data points get higher value of $\boldsymbol{\Gamma}$. When $\tau$ becomes larger, $\boldsymbol{\Gamma}$ turns to be more average. What is more, it indicates that our proposed method can also tackle the scenario when the initial masses $\boldsymbol{a}$ and $\boldsymbol{b}$ are not equal.

**Solving $\boldsymbol{\pi}^*$ of UOT/PUOT.** After we obtain the marginal probability of UOT/PUOT, we can therefore transfer UOT/PUOT into traditional optimal transport problem for finding the solution of $\boldsymbol{\pi}^*$ according to corollary 1. Specifically, we adopt OT solvers with $\ell_2$-norm regularization (Blondel et al., 2018) for finding the UOT mapping solution $\boldsymbol{\pi}^*$ by utilizing proposed transform coefficient method and the results are shown in Fig. 1(d). Apparently, it provides sparse and smooth matching among these normal data samples across domains. Meanwhile, we further adopt sparse OT solvers by setting 2-nonzero-elements per-column (Liu et al., 2023) for solving $\boldsymbol{\pi}^*$ on PUOT with learned marginal probability. The results are shown in Fig. 2(c) and as $\tau$ increases from 0.01 to 10, the number of matching pairs increases simultaneously which aligns with our expectations.

**Time consumption analysis.** We now provide an empirical evaluation of time consumption of the proposed method. We sample the same number of data samples (i.e., $m = n$) ranging from 100 to 1000 from $\boldsymbol{x} \sim \mathbb{P}_X$ and $\boldsymbol{z} \sim \mathbb{P}_Z$ for both UOT and PUOT respectively. We compare our methods with the following baselines: (1) Regularization Path (Chapel et al., 2021) algorithm which directly solves the UOT/PUOT problem with $\ell_2$-penalty. (2) Majorization-Minimization (MM) (Chapel et al., 2021) algorithm which solves the KL-penalized UOT/PUOT problem with multiplicative update. (3) Utilizing the following OT solvers, i.e., Sinkhorn (Cuturi, 2013), Smooth OT



(a) Experiments on $\eta = 10^0$ (b) Experiments on $\eta = 10^1$ (c) Experiments on $\eta = 10^3$ (d) Experiments on $\eta = 10^5$

Figure 5: The results of $\pi^*$ with different value of $\eta$ on CROT-Sinkhorn algorithm ($\epsilon = 0.1$).

with $\ell_2$-norm (Blondel et al., 2018) and Sparse OT with 2-nonzero-elements per-column (Liu et al., 2023) after calculating the determined marginal probability of UOT/PUOT. Note that the coefficients in front of the regularization term are all given as 0.1. We perform five random experiments and report the average results in Fig. 3(a)-(f) We can observe that although Regularization Path can obtain sparse and accurate results, it has highest computation cost than other methods. What is worse, the blue line in Fig. 3(d) is broken since it needs high overhead on storage space (more than 450G) for computation while we cannot satisfy the condition. Meanwhile our proposed transform coefficient approach with Sinkhorn is even more slightly efficient than MM Algorithm, indicating that our proposed approach is efficient in solving UOT/PUOT. Moreover, our proposed approach of utilizing transform coefficients can be seamlessly integrated with various OT solvers, enhancing flexibility and enabling the attainment of accurate results.

**Solving $\pi^*$ using CROT.** Then we investigate CROT by multiply the cost matrix $C$ with multipliers and a relatively large value of $\eta$. We sample 50 data ($M = N = 50$) from $\boldsymbol{x} \sim \mathbb{P}_X$ and $\boldsymbol{z} \sim \mathbb{P}_Z$ with uniform weights respectively. We measure pairwise cost $C_{ij}$ via square Euclidean distance and then divide by $\max_{ij}[C_{ij}]$ for normalization following (Flamary et al., 2021) and set $\eta = 10^5$ empirically. We first directly apply Sinkhorn (set $\epsilon = 0.1$ for entropy regularization term) for solving $\pi^*$ and the results have been shown in Fig. 4(a). Apparently, Sinkhorn just provides rather dense solution which cannot meet the actual needs. If we directly multiply the cost matrix with $\eta$, Sinkhorn will lead to null solution as shown in Fig. 4(b). To provide more sparse solution while using Sinkhorn, we first set $\tau = 10^5$ to transform OT into UOT for finding the multipliers $\boldsymbol{s}$ to build up CROT. Then we apply Sinkhorn on CROT and the results are shown in Fig. 4(c). We can observe that CROT with Sinkhorn provides relatively sparse solution on $\pi^*$. Meanwhile we further plot the heatmap and histogram of $\boldsymbol{s}$ as shown Fig. 4(c). We can observe that $\boldsymbol{s}$ shows a trend of long tail distribution, indicating that at least $80\%$ of non-matching pairs ($\pi_{ij} \to 0$) are exploited. It illustrates that utilizing $\boldsymbol{s}$ with CROT can reach more sparse solution via providing useful guidance beforehand.

Moreover, we calculate the discrepancy $e$ between the proposed matching solution $\pi^*$ and the OT solution $\pi^o$ learned by network-flow algorithm via $e = \sum_{i,j}[\|\pi_{ij}^* - \pi_{ij}^o\|]$. We sample the same number of data samples (i.e., $M = N$) ranging from 100 to 2000 from $\boldsymbol{x} \sim \mathbb{P}_X$ and $\boldsymbol{z} \sim \mathbb{P}_Z$ for calculation. We first directly solve the problem via Sinkhorn, Smooth OT with $\ell_2$-norm and Sparse OT with 2-

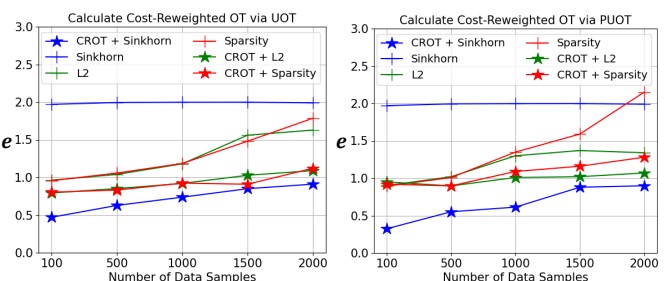

Figure 6: The discrepancy analysis on OT solution $\pi^o$ and solution $\pi^*$ obtained by CROT with multipliers $\boldsymbol{s}$ obtained from UOT/PUOT and different OT solvers.

nonzero-elements per-column. Meanwhile we conduct CROT using corollary 4 by transforming OT into UOT or PUOT to find multipliers $\boldsymbol{s}$ and then adopt different OT solvers on reaching $\pi^*$. We set $\tau = \eta = 10^5$ and the coefficients in front of the regularization term are all given as 0.1 empirically. The discrepancy results are shown in Fig. 6 and we can observe that original Sinkhorn algorithm has largest discrepancy indicates that it could easily provide inaccurate solutions. When the number of data samples increases, the discrepancy of current sparse methods (e.g., $\ell_2$-norm and Sparse OT) continues to grow. Meanwhile, all CROT-based methods achieve much better results than previous solutions even for $\ell_2$-norm and Sparse OT. Moreover CROT with Sinkhorn reaches the lowest value. It indicates that CROT with Sinkhorn is effective for obtaining the most accurate results of $\pi^*$ that other methods, regardless of whether calculating multipliers $\boldsymbol{s}$ on UOT or PUOT.

Table 1: Classification accuracy (%) on *Office-Home* for unsupervised domain adaptation

| Method | Ar→Cl | Ar→Pr | Ar→Rw | Cl→Ar | Cl→Pr | Cl→Rw | Pr→Ar | Pr→Cl | Pr→Rw | Rw→Ar | Rw→Cl | Rw→Pr | Avg |
|---|---|---|---|---|---|---|---|---|---|---|---|---|---|
| ResNet He et al. (2016) | 34.9 | 50.0 | 58.0 | 37.4 | 41.9 | 46.2 | 38.5 | 31.2 | 60.4 | 53.9 | 41.2 | 59.9 | 46.1 |
| DeepJDOT Damodaran et al. (2018) | 50.7 | 68.6 | 74.4 | 59.9 | 65.8 | 68.1 | 55.2 | 46.3 | 73.8 | 66.0 | 54.9 | 78.3 | 63.5 |
| JUMBOT Fatras et al. (2021) | 55.2 | 75.5 | 80.8 | 65.5 | 74.4 | 74.9 | 65.2 | 52.7 | 79.2 | 73.0 | 59.9 | 83.4 | 70.0 |
| JUMBOT + UOT(Sparse) | 57.1 | 76.9 | 81.6 | 66.1 | 74.7 | 75.5 | 66.0 | 53.4 | 79.6 | 74.2 | 60.3 | 83.7 | 70.8 |
| JUMBOT + UOT(CROT + Sparse) | **57.8** | **77.2** | **82.3** | **66.7** | **76.1** | **75.9** | **66.8** | **53.9** | **80.7** | **75.5** | **61.0** | **84.6** | **71.5** |

**Tuning on hyperparameter $\eta$.** Last but not least, we investigate the effects on choosing different value of $\eta$. We vary $\eta$ in range of $\{10^0, 10^1, 10^3, 10^5\}$ on CROT with 50 samples ($M = N = 50$) and reported in results in Fig. 5. Since the magnitude of $s$ is too small, it cannot provide significant effect when $\eta$ is rather small for obtaining sparse $\pi^*$. Therefore, it is also essential to multiple $s$ with a larger value $\eta$ to further enhance the impact of the multipliers for obtaining sparse results.

**UOT application for Unsupervised Domain Adaptation (UDA).** We demonstrate the application on using UOT for Unsupervised Domain Adaptation (UDA) where the goal is to assign labels to the unlabeled target domain data using the labeled source domain data. Meanwhile, the unlabeled target domain data shares the same class categories as the labeled source domain data (Agarwal et al., 2021; Flamary et al., 2016; Redko et al., 2017; Nguyen et al., 2021). We follow the same framework and experimental settings as UDA model Joint Unbalanced MiniBatch OT (JUMBOT) (Fatras et al., 2021). In our approach, we replace the entropy-based minibatch UOT component in JUMBOT with our proposed sparse UOT method, which incorporates UOT transform coefficient and $\ell_2$-norm OT solver for sparsification (Blondel et al., 2018). The regularization parameter for the $\ell_2$-norm term is set to 0.1. We then set $\tau = 0.5$ for KL divergence term in UOT and proceed to conduct experiments on the *Office-Home* datasets (Venkateswara et al., 2017). *Office-Home* is a benchmark for visual domain adaptation, which consists of 15,500 images in 65 object classes in office and home settings with four dissimilar domains: Artistic images (**Ar**), Clip Art (**Cl**), Product images (**Pr**) and Real-World (**Rw**). We report the average classification accuracy for various tasks on *Office-Home* in Table 1. From that we can observe that our proposed UOT with transform coefficient and $\ell_2$-norm sparse OT solver can provide better results than JUMBOT on UDA scenario with real data. This approach is justified as sparse mapping helps to eliminate ambiguous transportation plans, while simultaneously offering more reliable and robust solutions. Moreover, we further utilize CROT with $\ell_2$-norm sparse OT solver in solving UOT and it achieves the best performance, indicating the proposed method provides more accurate results via considering the useful guidance of multipliers.

## 5  RELATED WORKS

**Unbalanced Optimal Transport.** UOT with KL divergence has been widely investigated for dealing with diverse applications (Peyré et al., 2019; De Plaen et al., 2023; Séjourné et al., 2019). Different types of UOT solutions can be distinguished by whether or not they incorporate an entropy regularization term. Involving entropy in UOT can enhance the model scalability but results in dense matching results (Sinkhorn & Knopp, 1967; Balaji et al., 2020). Latest, (Chapel et al., 2021) further considers UOT without entropy terms by Majorization-Minimization (MM) (Chizat et al., 2018; Sun et al., 2016) or regularization path methods (Mairal & Yu, 2012; Massias et al., 2018; Liu & Nocedal, 1989). However, the nature of MM algorithm inherits inexact proximal point term (Xie et al., 2020) and thus it cannot overcome the defects of entropy and leading to dense mapping when $\tau$ becomes larger. Meanwhile regularization path methods could be relatively slow in computation especially when $\tau \rightarrow +\infty$. Furthermore, as the number of samples increases, it can lead to high storage space consumption which can be problematic. Therefore, how to efficiently provide sparse and accurate solution on both UOT and PUOT is still a challenging problem.

## 6  CONCLUSION

In this paper, we first propose transform coefficients method to determine the marginal probability of UOT/PUOT. It reveals that the essence of UOT/PUOT is to reweight different samples with its pairwise distance accordingly. Then we can directly transform UOT/PUOT into classical OT problem via proposed UOT/PUOT transform coefficients. Meanwhile we can further utilize the obtained multipliers when calculating transform coefficients to establish Cost-Reweighted OT (CROT) to obtain more sparse and accurate transportation plan. We also conduct numerical and real data experiments to validate the efficacy of our proposed methods. In conclusion, the transform coefficients approach provides a complete new insight on connecting UOT, PUOT and OT problems.

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
