# OpenReview forum: "Solving (partial) unbalanced optimal transport via transform coefficients and beyond"
_ICLR.cc/2024/Conference — Submitted to ICLR 2024_

### Official Review · Reviewer_ZcUR · 2023-10-26

**Soundness:** 2 fair
**Presentation:** 3 good
**Contribution:** 1 poor
**Rating:** 3
**Confidence:** 4

**Summary:**

Summary: The authors propose a new method called the transform coefficient method for solving UOT with KL divergence without entropy regularization term. This is done by making calculations based on the KKT conditions of UOT and finding the proposed transform coefficients for determining the marginal probability distributions.

This paper also proposed Cost-Reweighted Optimal Transport (CROT) for achieving a more sparse and accurate OT matching solution.

**Strengths:**

Strengths:
This paper offers a way to solve the unregularized UOT problem. In addition, since it is unregularized it will not return dense solutions, unlike entropic-regularized methods.

**Weaknesses:**

Weaknesses:
- Many results of this paper are either marginal derivations of past works or duplicated results. For example, Corollary 4 is a weaker result of Theorem 13 in [1] where the former only has asymptotic convergence. At the same time, the latter provides a non-asymptotic convergence rate in terms of tau. I believe that even for PUOT obtaining the convergence rate in terms of tau would not be too difficult.
- While CROT indeed yields sparsity compared to an entropic-regularized transport plan, an $\ell_2$-regularized transport plan would also induce sparsity while giving an accelerated convergence rate.
- There is no complexity analysis for algorithm 1. Judging by the numerical experiments, I believe that the proposed method will be slower than the entropic-regularized Sinkhorn (which is expected since the entropic regularization induces acceleration) or the gradient extrapolation method in [1].
- The theoretical contribution is quite limited, as the authors only use KKT conditions to derive the transform coefficients.

References
[1] "On Unbalanced Optimal Transport: Gradient Methods, Sparsity and Approximation Error".
Quang Minh Nguyen, Hoang Huy Nguyen, Lam Minh Nguyen, Yi Zhou

**Questions:**

Questions:
Does this paper require the cost matrix to be $\ell_2$ Euclidean distance?

---

### Official Review · Reviewer_x5ni · 2023-10-31

**Soundness:** 2 fair
**Presentation:** 1 poor
**Contribution:** 2 fair
**Rating:** 3
**Confidence:** 4

**Summary:**

In this paper, the authors propose a method named "transform coefficients" for solving the (partial) unbalanced optimal transport (UOT) where the marginal constraints are penalized by using the Kullback-Leibler divergence. In particular,

1. They first claim (without any rigorous proof) that the marginals of the optimal solution $\boldsymbol{\pi^*}$ of the UOT problem can be computed via terms called "transform coefficients" without calculating $\boldsymbol{\pi^*}$ (see Proposition 1);

2. Next, they leverage the method of Lagrange multipliers to derive updates for those transform coefficients $\boldsymbol{\varphi},\boldsymbol{\delta}$ and the Lagrange multiplier $\boldsymbol{s}$. Then, they obtain the marginals of the optimal solution based on the results of Proposition 1;

3. Given those marginals, they rewrite the UOT problem as a OT problem with a new cost matrix $\widetilde{\boldsymbol{C}}$ where $\widetilde{\boldsymbol{C}}$ is the element-wise product of the UOT cost matrix $C$ and a factor $\eta$ of Lagrange multiplier $\boldsymbol{s}$. By setting sufficiently large values for $\eta$, they obtain a sparse optimal solution $\boldsymbol{\pi^*}$.

Finally, they run some numerical experiments to justify their proposed method.

**Strengths:**

1. The paper proposes a new approach to solve the (partial) unbalanced optimal transport (UOT) problem.
2. Several experiments are conducted to justify the effectivenes of the proposed method.

**Weaknesses:**

1. All the results presented in the paper are not associated with rigorous theoretical guarantee, which makes the results less reliable.

2. There are many claims without justification in the paper. For instance, in the paragraph above Proposition 3, the authors claimed that the solution $\boldsymbol{\pi^*}$ should be sparse in most cases without any explanations and citing relevant papers.

3. The name of partial unbalanced optimal transport is confusing. In the literature [1], this problem is referred to as semi-constrained optimal transport. On the other hand, when talking about partial transportation, ones think about the problem of transporting a fraction of the mass as cheaply as possible consdiered in [2, 3].

4. The writing of the paper is not good. There are many undefined notations (e.g. KL divergence, $\widehat{C}^*_{i,j}$ in Proposition 1), grammatical errors as well as typos (see Question section). Furthermore, the authors also need to cite more relevant papers.

**References**

[1] Khang Le, Huy Nguyen, Quang Nguyen, Tung Pham, Hung Bui, Nhat Ho. On Robust Optimal Transport: Computational Complexity and Barycenter Computation. In NeurIPS, 2021.

[2] Laetitia Chapel, Mokhtar Z. Alaya, Gilles Gasso. Partial Optimal Transport with Applications on Positive-Unlabeled Learning. In NeurIPS, 2020.

[3] Khang Le, Huy Nguyen, Tung Pham, Nhat Ho. On Multimarginal Partial Optimal Transport: Equivalent Forms and Computational Complexity. In AISTATS, 2022.

**Questions:**

1. The authors should cite more relevant papers:
- In the Introduction section, the papers [1, 2] should be cited for the references of optimal transport problem.
- In the last paragraph of Section 1, the authors should cite papers which previously considered the partial unbalanced optimal transport problemm namely [3]. Moreover, it is also necessary to explain how the term 'partial' in that problem differs from the one in the context of partial transportation problems in [4, 5].

2. What is the stop criterion for Algorithm 1? Given that stop criterion, how do we know the output transform coefficients and Lagrange multiplier are good enough?

3. In equation (3), does the equality $\boldsymbol{a}^{\top}\mathbf{1}_M=\boldsymbol{b}^{\top}\mathbf{1}_N$ necessarily hold true?

4. How are the notations $\widehat{C}^*_{i,j}$ defined in Proposition 1?

5. Are there any theoretical guarantee for the propositions and corollaries in the paper? If yes, it should be at least in the supplementary material.

6. Why the optimal solution $\boldsymbol{\pi^*}$ should be sparse in most cases?

7. What is the computational complexity of Algorithm 1?

8. In the cost-reweighted optimal transport problem, when the value of $\eta$ increases, then how far is the optimal solution of this problem from that of the UOT problem in equation (2)?

9. Grammatical errors:
- In the 'Brief Summary' paragraph below Corollary 2, 'different value of $\tau$' --> 'values'
- In the beginning of Section 3.2, 'exist optimization problem' --> 'existing'
- Below Proposition 3, 'If $s_{ij}$ is much approaches to 0' --> 'If $s_{ij}$ approaches zero'

10. Two terms 'KL divergence' and 'KL Divergence' are both used in the paper, which makes the paper inconsistent.

**References**

[1] C. Villani. Optimal Transport: Old and New, Volume 338. Springer Berlin Heidelberg, 2009.

[2] C. Villani. Topics in Optimal Transportation. American Mathematical Society, 2003.

[3] Khang Le, Huy Nguyen, Quang Nguyen, Tung Pham, Hung Bui, Nhat Ho. On Robust Optimal Transport: Computational Complexity and Barycenter Computation. In NeurIPS, 2021.

[4] Laetitia Chapel, Mokhtar Z. Alaya, Gilles Gasso. Partial Optimal Transport with Applications on Positive-Unlabeled Learning. In NeurIPS, 2020.

[5] Khang Le, Huy Nguyen, Tung Pham, Nhat Ho. On Multimarginal Partial Optimal Transport: Equivalent Forms and Computational Complexity. In AISTATS, 2022.

---

### Official Review · Reviewer_8FWU · 2023-11-01

**Soundness:** 1 poor
**Presentation:** 2 fair
**Contribution:** 2 fair
**Rating:** 1
**Confidence:** 4

**Summary:**

This work proposed the transform coefficient method that aims to equivalently transform the Unbalanced Optimal Transport (UOT) or Partial UOT (PUOT) problems into OT from which standard solvers can be used. Experiments were then provided.

**Strengths:**

The way of doing reweighting in this paper is somewhat new and a bit interesting. It can be potential future direction, if more careful analysis and study are given to treat the methodology.

**Weaknesses:**

1) The literature review and thus positioning of this work's contribution is very insufficient. Please cite the relevant work as mentioned below and properly re-write the relevant literature discussion. E.g:
 - "Previous researches always add entropy regularization term for solving OT, UOT and PUOT. " --> See [1] for UOT and [2] for OT both with l2-regularization.
- "As we already have abundant methods available for tackling OT problems, the idea of converting UOT to OT brings us a brand new perspective in dealing with UOT." --> Results like this were done in the literature, where [1] showed tight bounds on how fast UOT converges to OT with the growth of $\tau$. So this is not the first work.

2) This paper lacks mathematical rigor in the formulation and development of the methods. The idea is presented in a very heuristic form.
- Proposition 1 is given using the notion of "transformed pairwise distance" that is never properly introduced in the paper. Maybe  solving the optimal "transformed pairwise distance" might correspond to solving the original UOT problem, thereby invalidating the whole purpose of this paper. Other results are also very informal.
- Following up the above comment, the Algo 1 in the paper runs iteratively to adjust the "transformed pairwise distance" (I know it's $s_{ij}$, but it is equivalent way of saying), but we can't know how long it should run to converge.
- Following up the above comment, you in fact do not equivalently yet approximately convert the UOT/PUOT problem into OT. Moreover, we cannot know how good such approximate conversion is. So your whole framework and idea, though interesting, are invalid. Please note that even if you use OT solver to solve your converted OT problem, how would you next retrieve your UOT solution out of that because your equivalence is approximate in the first place?

3) Experiments are on synthetic data only, and thus are very weak.


[1] Nguyen, Quang Minh, Hoang Hai Nguyen, Yi Zhou and Lam M. Nguyen. “On Unbalanced Optimal Transport: Gradient Methods, Sparsity and Approximation Error.” (2022).
[2] Essid, Montacer and Justin M. Solomon. “Quadratically-Regularized Optimal Transport on Graphs.” SIAM J. Sci. Comput. 40 (2017): A1961-A1986.

**Questions:**

See Weaknesses

---

### Official Review · Reviewer_bjot · 2023-11-02

**Soundness:** 3 good
**Presentation:** 3 good
**Contribution:** 2 fair
**Rating:** 3
**Confidence:** 4

**Summary:**

The paper addresses the solving of Unbalanced Optimal Transport (UOT) and Partial Unbalanced Optimal Transport (PUOT) with KL divergences that relax the usual marginal constraints of OT and without entropic regularization. Departing from the common scheme that estimates the related coupling matrix, the paper proposes the so-called transform coefficients based on the KKT optimality conditions of the UOT and PUOT optimization problems. Using these coefficients, iterative algorithms are proposed to estimate first the marginal distributions and the Lagrange multipliers of the corresponding problems. In a second stage a Cost-Reweighted OT (CROT) problem is formulated with the estimated marginals and with a modified cost matrix that takes into account the multipliers in order to compute the sought coupling matrix. Numerical experiments illustrates the estimated marginals, the computation timings, the quality of estimated coupling matrix by CROT on simulated datasets. Application of the method for unsupervised domain adaptation highlights the effectiveness of the approach.

**Strengths:**

- The paper is well written. It  reviews the background and main works on solving UOT and PUOT problems.  The rationales behind the proposed transform coefficients and the two-stage procedure to solve UOT and PUOT are clearly stated.
- The main originality of the work resides in revisiting UOT and PUOT via the transform coefficients that allow to compute the modified marginal distributions due to the marginal constraints relaxation without solving for the coupling matrix $\pi$. The proposed algorithm not only outputs the transform coefficients that allow to compute the modified marginals but also the Lagrange multipliers $\boldsymbol{s}$ associated to the positivity constraint on $\pi$. $\boldsymbol{s}$ provides information of the sparsity pattern of $\pi$. This is exploited by the Cost-Reweighted OT (CROT) problem which computes the coupling matrix of the UOT/PUOT problem.
- Empirical evaluations show that the proposed method provides a computation gain compared to competitors (Majorization-Minimization algorithm, regularization path). Also numerical experiments show that the method is effective on domain adaptation task in comparison with methods such as DeepJDOT, JUMBOT.

**Weaknesses:**

- Convergence guarantees of Algorithm 1 are not exposed. The used convergence criterion should be stated and should be related to the derived KKT conditions. The same remark holds for PUOT related algorithm.
- Formal proofs or elements of proof of the theoretical results on which CROT algorithm is based are not proposed. Also theoretical guarantees on the discrepancy between the estimated $\pi^*$ by CROT and the true coupling matrix corresponding to the UOT are not provided.
- The motivation of solving the reweighted OT using Sinkhorn, OT with L2-norm or sparse OT algorithm is a bit hazy. Moreover those solvers necessitate hyper-parameters that are not easy to tune.  As such the obtained coupling matrix $\pi$ may highly differ from the one that can be achieved while solving directly the UOT/PUOT problem using existing methods.
- The reported empirical evaluations are not always convincing. For instance, when assessing the quality of $\pi^*$ by CROT, the distance $\sum_{ij} |\pi^*_{ij} - \pi^o_{ij}|$ where $\pi^o$ is the solution of a plain OT (according to my understanding) computed by network-flow algorithm. It would be fair and meaningful to contrast $\pi^*$ with the coupling matrix given by existing UOT or PUOT solvers. Another example is Figure 4 which illustrates $\pi^*$ on a UOT problem with $\tau=10^5$. Such setting will enforce the marginal constraints to be met, leading to a coupling solution close to OT.  Therefore, it is unclear why Figure 4 does consider Sinkhorn instead of non-regularized OT.
- In the domain adaptation task, Table 1 does not consider JUMBOT coupled with UOT solved by existing methods in order to provide a fair evaluation and assess the benefit of CROT over existing UOT solvers on this specific application.
- Authors should proof read the paper for some inconsistent text formulations.

**Questions:**

- The derivation of transform coefficients is based on the assumption that $||\boldsymbol{a}||_1 = ||\boldsymbol{b}||_1$. In some UOT cases this assumption may not hold true. How the method behaves in such setting?
- The Lagrange parameters $\boldsymbol{s}$ provides the sparsity pattern of $\pi$. Instead of introducing the modified cost matrix $\tilde{ \boldsymbol{C}}$ and the extra hyper-parameter $\eta$, why the paper does not solve directly an OT problem with some entries of $\pi$ fixed to null values according to the sparsity map encode by $\boldsymbol{s}$?
- Theoretically, how close is the estimated coupling matrix $\pi^*$ by CROT to the true coupling matrix corresponding to the UOT problem?
- In the timing evaluation another baseline method to be considered is the MM method for L2-penalty along with the regularization path.

---

### Official Review · Reviewer_vADR · 2023-11-06

**Soundness:** 2 fair
**Presentation:** 2 fair
**Contribution:** 2 fair
**Rating:** 3
**Confidence:** 4

**Summary:**

This paper proposes a method for the calculation of unbalanced and partially-balanced optimal transport maps. i.e. transport maps where the source and target masses are not equal.

The authors consider a special case of Eq. (2.4) of Chizat et al. (2018) where the divergence function is the KL divergence. They analyze the KKT conditions for this problem and propose an iterative algorithm for obtaining the marginal probabilities of the solution. Then, these marginal probabilities can be plugged into a standard (balanced) OT solver.

**Strengths:**

* The specific approach outlined here seems new.
* There could be cases where a sparse solution is required (though I am not sure which). In that case the approach outlined in the paper will satisfy the requirement.

**Weaknesses:**

* This subject has been explored in greater generality (e.g. in Chizat et al. 2018) and algorithms for computing unbalanced optimal transport have already been proposed and used. The unique contribution here is that the resulting transport map is sparse. However, the benefit of that is not clear to me.
* There is no analysis of the convergence speed of Algorithm 1.
* The approach seems to scale poorly and accordingly the experimental data sets are all very small. The iterations of Algorithm 1 are O(NM), as expensive as Sinkhorn iterations, and that's before the main step of solving the OT problem.
* The text needs to be thoroughly copy-edited as it contains numerous errors. e.g. "it does not match most of situations", "for avoid heavy computations", "if s_i,j is much approaches to 0", etc.
* The text in the figures in tiny. Figure 3 is unreadable when printing in black&white.

**Questions:**

* What is the convergence speed of Algorithm 1?
* Can you find specific use cases where a sparse OT map is absolutely necessary?

---

### Meta-Review · Area_Chair_igoP · 2023-12-07

**Metareview:**

The authors consider (partial) unbalanced optimal transport problem. The authors propose transform coefficient method based on KKT optimal condition, then the authors propose iterative algorithm to estimate marginal distributions and Lagrange multipliers, after that the authors solve the Cost-Reweighted OT (CROT) problem. The authors show the advantages of the proposed method on several numerical experiments.

The proposed approach is interesting. However, the reviewers raised several issues about the proposed approach (e.g., convergence, scalability, the goodness of the solution, retrieve the solution of the original (P)UOT, stopping condition, etc). Overall, the submission is not ready for publication yet. I urge the authors to incorporate the reviewers' comments to improve the submission.

**Justification For Why Not Higher Score:**

Several issues are raised by the reviewers about the proposed algorithm. However, there is no rebuttal from the authors, those raised issues still remain. So, I think the submission is not ready for publication yet.

**Justification For Why Not Lower Score:**

N/A

---

### Decision · Program_Chairs · 2024-01-16

Reject